# Effect of Evening Primrose Oil Supplementation on Biochemical Parameters and Nutrition of Patients Treated with Isotretinoin for Acne Vulgaris: A Randomized Double-Blind Trial

**DOI:** 10.3390/nu14071342

**Published:** 2022-03-23

**Authors:** Agnieszka Kaźmierska, Izabela Bolesławska, Paweł Jagielski, Adriana Polańska, Aleksandra Dańczak-Pazdrowska, Grzegorz Kosewski, Zygmunt Adamski, Juliusz Przysławski

**Affiliations:** 1Department of Bromatology, Poznan University of Medical Sciences, 60-806 Poznan, Poland; a.kazmierska@interia.pl (A.K.); grzegorzkosewski@ump.edu.pl (G.K.); jprzysla@ump.edu.pl (J.P.); 2Department of Nutrition and Drug Research, Institute of Public Health, Faculty of Health Sciences, Jagiellonian University Medical College, 31-066 Krakow, Poland; paweljan.jagielski@uj.edu.pl; 3Laboratory of Connective Tissue Diseases, Department of Dermatology and Venereology, Poznan University of Medical Sciences, 60-355 Poznan, Poland; apolanska@ump.edu.pl; 4Department of Dermatology, Poznan University of Medical Sciences, 60-355 Poznan, Poland; aleksandra.danczak-pazdrowska@ump.edu.pl (A.D.-P.); zadamski@ump.edu.pl (Z.A.)

**Keywords:** acne, isotretinoin, evening primrose oil, lipid profile, aminotransferase levels

## Abstract

Background: Acne vulgaris is one of the most common skin diseases. One of the therapeutic options recommended for severe acne or acne that has not responded to previous therapies is isotretinoin. However, its use may lead to adverse changes in the serum lipid profile and increased levels of transaminases. In this study, we evaluated the effect of supplementation with evening primrose oil in acne vulgaris patients treated with isotretinoin on blood lipid parameters and transaminase activity. Methods: Study participants were randomly assigned to two treatments: conventional with isotretinoin (25 patients) and novel with isotretinoin combined with evening primrose oil (4 × 510 mg/day; 25 patients) for 9 months. Results: Compared to isotretinoin treatment, isotretinoin treatment combined with evening primrose oil had a positive effect on TCH concentrations (mean: 198 vs. 161, *p* < 0.001), LDL (95.9 vs. 60.2, *p* < 0.001), HDL (51.0 vs. 48.0, *p* < 0.001), TG (114 vs. 95.0, *p* < 0.001), ALT (24.0 vs. 22.0, *p* < 0.001), and AST (28.0 vs. 22.0, *p* < 0.001), but had no effect on the energy and ingredient content of the diets (*p* > 0.05) after treatment. Conclusion: Evening primrose oil was found to have beneficial effects on lipid profiles and transaminase activity during isotretinoin treatment. However, longer studies are needed to make more reliable decisions regarding the use of evening primrose oil and its safety in clinical practice. The evening primrose oil treatment group also showed a reduction in dietary energy due to a reduction in dietary protein and carbohydrates.

## 1. Introduction

Acne vulgaris, a chronic inflammatory disease of the sebaceous glands [1], is one of the most common skin diseases [2,3,4]. The incidence of acne vulgaris is highest during adolescence, but it can also occur at other times in life [2,4,5].

The primary mechanism of the disease involves excessive sebum secretion, keratinocyte hyperproliferation in the hair follicle, altered bacterial colonization, and a host inflammatory response [4,5,6]. The exact sequence of these events is not clear, but the main pathophysiological factor is probably an androgen-induced increase in sebum production and secretion, combined with qualitative changes in sebum [5]. Characteristic changes in sebum composition in patients with acne include decreased linoleic acid, increased sapienic acid, squalene, diacylglycerols (DG), and lipid peroxides, and an increased ratio of saturated to unsaturated fatty acids [7,8,9].

The occurrence of acne is associated with profound negative effects on mental health, including increased rates of mood disorders, psychiatric hospitalizations, school absenteeism, job loss, and suicide [10].

Treatment of acne depends on its form and severity. The mainstay of acne therapy is retinoids, which, depending on the form of the disease and its severity, can be used generally or systemically [11,12,13,14]. In light of recent studies, isotretinoin is recommended as a first-line treatment for moderate to severe inflammatory acne [2,11,15,16,17,18,19].

However, the use of isotretinoin may lead to a number of adverse effects, which include changes in the serum lipid profile, including total cholesterol (TCH), cholesterol in the LDL fraction (LDL), and triacylglycerols (TG) [20,21], as well as an increase in aminotransferases [22,23,24].

Studies have shown that evening primrose oil (EPO) supplementation may be effective in improving lipid profiles, but results have been inconsistent. For example, oral intake of EPO at a dose of ≤4 g/day was found to significantly decrease serum TG levels and significantly increase HDL levels in hyperlipidemic subjects [25].

The biological effect of evening primrose oil is due to its composition and the biological properties of its components. The most important, in terms of quantity, are polyunsaturated fatty acids (PUFAs), mainly linoleic acid (LA) and γ-linolenic acid (GLA), which belong to the group of omega-6 acids [26,27].

The available literature shows that linoleic acid lowers blood cholesterol by 13.8%, as predicted by the Keys equation [28], while γ-linolenic acid lowers TG levels and raises HDL levels [29]. In an animal model, diets containing γ-linolenic acid from evening primrose oil also significantly reduced serum TG and TCH levels compared to other oils (palm and safflower) [30]. A review by Hooper et al. [31] confirmed that increased intake of omega-6 fats reduces serum total cholesterol levels without affecting the other fractions.

The beneficial effect of evening primrose oil in combination with hemp oil and a hot-nature diet on liver enzyme levels was also demonstrated [32]. Higher dietary linoleic acid levels were also generally associated with lower serum liver enzyme levels [33].

Guided by the above rationale, a study was undertaken to evaluate the effect of evening primrose oil supplementation in acne vulgaris patients treated with isotretinoin on blood lipid parameters and transaminase activity and diet.

## 2. Materials and Methods

The study was designed as a randomized trial according to CONSORT standards (see Appendix A) [34,35] and conducted according to the guidelines of the Declaration of Helsinki. The study protocol was approved by the Bioethics Committee of the Poznan University of Medical Sciences (ref 268/15). All participants received information about the study. All subjects were aware that their participation was voluntary and they could withdraw at any time without giving reasons.

### 2.1. Study Participants

A group of 50 patients aged 18 to 30 years (mean age 22.0 ± 2.07 years) with a diagnosis of moderate to severe acne vulgaris participated in the study after giving written consent. They were randomly divided into two groups: 25 subjects for the IOW study group (mean age 22.5 ± 1.92 years) and 25 subjects for the control group I (mean age 21.6 ± 2.14 years). There was no significant difference in age between groups (*p* = 0.116). Patients were eligible for the study on the basis of dermatological assessment. The inclusion criteria were a diagnosis of moderate to severe acne or lack of improvement after previous medication. Exclusion criteria included pregnancy or breastfeeding, metabolic diseases, liver failure, elevated blood cholesterol levels, use of statins, high vitamin A levels in the body, and hypersensitivity to vitamin A. Potential participants were recruited from patients treated at the outpatient clinic of the Department of Dermatology, Poznan University of Medical Sciences. After consenting to participate in the study, they were examined by a physician during the inclusion visit to meet protocol requirements.

### 2.2. Study Design

The study used randomization in a simple procedure (computerized random numbers). An observational study was carried out in patients treated at the outpatient clinic of the Department of Dermatology, Poznan University of Medical Sciences. The decision to introduce isotretinoin according to dermatological guidelines was made by experienced dermatologists.

Study participants were randomly assigned (allocation ratio: 1:1) to arms I and IOW. In arm I, an individually determined dose of oral isotretinoin (10 to 40 mg isotretinoin) was administered by a dermatologist. In arm IOW, subjects received isotretinoin plus encapsulated evening primrose oil (Oenothera paradoxa) in the amount of 2 capsules (2 × 510 mg) in the morning and 2 capsules (2 × 510 mg) in the evening. The duration of the intervention was 9 months and concurrently ran in both arms. All subjects took an individually determined dose of oral isotretinoin (10 to 40 mg of isotretinoin) by a dermatologist. Treatment was usually started at a dose of 0.5 mg/kg body weight/day. The therapeutic dose ranged from 0.5–1 mg/kg body weight/day. For isotretinoin, the cumulative dose for the entire treatment period was 120–150 mg/kg body weight. Of the 50 subjects included in the study, all completed 9 months of treatment (group I 25 subjects, group IOW 25 subjects). No side effects or adverse events were reported. The study analyzed the results of biochemical parameters at baseline and after 9 months (although patients were followed up more frequently by a dermatologist). Dietary assessment (dietary history) was performed twice in study participants before and after the study. Biochemical parameters—total cholesterol (TCH), LDL fraction cholesterol (LDL), HDL fraction cholesterol (HDL), triglycerides (TG), aspartate transaminase (ASP), and alanine transaminase (ALT) levels—were also assessed twice.

The encapsulated evening primrose oil used in the study was extracted from Oenothera paradoxa and manufactured by Adamed Consumer Healthcare S.A. The capsule contained evening primrose seed oil (Oenothera paradoxa) at 510 mg and unsaturated fatty acids at 390 mg, including: linoleic acid (LA) at 347 mg, gamma-linolenic acid (GLA) at 42.4 mg, gelatin, and humectants glycerol and sorbitol. According to the manufacturer’s recommendations, the recommended intake is not exceeded, which is for adolescents and adults 1–2 capsules 2 times a day. Composition of evening primrose oil capsule is shown in Table 1.

During the intervention, participants were instructed not to make changes to their previous dietary and physical activity habits. In addition, during the intervention, participants were supervised by telephone by a dietician to verify adherence to the study protocol, especially the regular intake of oil capsules. To verify adherence to the study protocol, participants regularly returned empty oil containers to the research team. The primary outcomes of the study were to compare the effects of treatment of acne vulgaris with isotretinoin or isotretinoin combined with evening primrose oil on lipid profiles and aminotransferase levels. Here, we present the results of endpoints such as TCH, LDL, HDL, TG, and ALT and AST levels. All outcome measurements were collected at the University of Poznan Medical School before and after each intervention period. Parameters related to aminotransferase levels and lipid profiles were analyzed at ALAB Laboratory (Poznań, Poland). All results were assessed by the same methods for both arms.

Randomization was performed by an independent researcher using computer software (Excel, Microsoft Corp, Redmond, WA, USA). The randomization list was generated and the allocation order was concealed until allocation to the intervention. Neither study participants nor research staff knew the allocation order.

The minimum sample size was calculated using Statistica StatSoft 13.3 data analysis software based on the following assumptions: a Type I probability of error of α = 0.05 and an allocation ratio of 1:1. As assumed, the minimum sample size was 23 subjects, but assuming a 10% dropout rate, we assumed that at least 25 subjects should be recruited for the study. We calculated the minimum sample size based on changes in TCH previously reported by Acmaz et al. [36], which were before isotretinoin treatment (133.2 ± 20.3 mg/dL) and after treatment (153.3 ± 20.3 mg/dL). The power for this size is 0.9072.

### 2.3. Experimental Procedure

#### 2.3.1. Dietary Assessment

The assessment of nutrient intake was based on a dietary intake interview conducted according to the guidelines of the Institute of Food and Nutrition (IŻŻ) in Warsaw [37]. The study was conducted systemically, i.e., the interview on the intake of products and foods was conducted with each respondent for a period of 3 days before the start of treatment and 3 days after 9 months of treatment. The analysis of the questionnaire results concerning the qualitative and quantitative composition of whole food rations was performed using computer databases prepared on the basis of “Tables of composition and nutritional value of food” [38]. The assessment of the level of intake was carried out on the basis of an application prepared in the Diet 6.0 program [39].

#### 2.3.2. Biochemical Tests

From each fasting patient, 12 h after the last meal, 10 mL of venous blood was collected from an elbow vein puncture. Roche COBAS INTEGRA 400+ analyzer was used for determination of total cholesterol TCH, HDL, and TG.

##### Determination of total cholesterol (TCH) in blood serum

An enzymatic and colorimetric method was used to determine total cholesterol, in which cholesterol esterase (CE) hydrolyzed cholesterol esters to free cholesterol and fatty acids. Cholesterol oxidase (CHOD) then catalyzed the oxidation of cholesterol to the cholest-4-en-3-one form and hydrogen peroxide. In the presence of hydrogen peroxide peroxidase (POD), oxidative coupling of phenol and 4-aminoantipyrine (4-AAP) occurred with the formation of red colored quinine. The intensity of the resulting color, directly proportional to the concentration of cholesterol in the sample, was determined by measuring the absorbance at 512 nm.

##### Determination of HDL-cholesterol in blood serum

A homogeneous colorimetric enzymatic method was used to determine serum HDL-cholesterol concentration.

HDL-cholesterol concentration was enzymatically determined using cholesterol esterase and cholesterol oxidase attached from PEG to amino groups. Cholesterol esters were quantitatively broken down into free cholesterol and fatty acids under the influence of cholesterol esterase. In the presence of oxygen, cholesterol was oxidized by oxidase to 4-Δ-cholestenone and hydrogen peroxide. In this method, water-soluble complexes with lipoprotein particles LDL, VLDL, and chylomicrons were formed in the presence of magnesium sickle cell and dextran sulfate and were resistant to polyethylene glycol-modified enzymes.

The intensity of the resulting blue quinonimine dye, directly proportional to the concentration of HDL cholesterol in the sample, was determined by measuring the increase in absorbance at a wavelength of 583 nm.

##### Calculation of LDL fraction cholesterol

The cholesterol concentration of the low-density lipoprotein (LDL) fraction was determined using the calculation method proposed by Friedewald et al. [40]. It assumes that the concentration of LDL cholesterol is the difference between the concentration of total cholesterol and the sum of the concentration of HDL fraction cholesterol and triacylglycerols.

This is represented by the following formula:LDL (mg/dL) = TCH − HDL − (TG/5)
LDL (mmol/L) = TCH − HDL − (TG/2,2)
LDL − LDL – cholesterol
TCH − total cholesterol
HDL − HDL – cholesterol
TG − triglycerides

When serum triacylglycerols exceeded 4.00 mmol/L, in massive hypertriglyceridemia and in hyperlipoproteinemia (HLP) type III, a direct method was used [41].

##### Determination of serum triglycerides (TG)

A homogeneous colorimetric enzymatic and colorimetric (GPO/PAP) method with glycerophosphate oxidase and 4-aminophenazone was used. In this method, triacylglycerols were hydrolyzed by lipoprotein lipase (LPL) to glycerol and fatty acids. The glycerol, in a reaction catalyzed by glycerol kinase (GK), was then phosphorylated to glycerol-3-phosphate with ATP. Oxidation of glycerol-3-phosphate was catalyzed by glycerophosphate oxidase (GPO) with the formation of dihydroxyacetone phosphate and hydrogen peroxide. In the presence of POD peroxidase, oxidative coupling of 4-chlorophenol and 4-aminophenazone occurred under the influence of hydrogen peroxide with the formation of red-colored quinonimine, measured at 512 nm. The increase in absorbance was directly proportional to the concentration of triglycerides in the sample.

##### Determination of serum ALT and AST

For the determination of ALT activity, an immersion assay for quantification of specific ALT activity (GPT) was used with a Reflotron^®^ reflectance photometer from Roche. For the determination of aspartate transaminase (AST) activity, an optimized modified method based on the recommendations of the International Federation of Clinical Chemistry (IFCC) was used, which employs the Karmen–Bergmeyer technique of coupling malate dehydrogenase (MDH) and the reduced form of nicotinamide adenine dinucleotide (NADH) during the detection of AST compound in serum [42,43]. The ratio of change in absorbance at light wavelengths of 340 nm and 405 nm caused by conversion of the NADH form to NAD+ is directly proportional to the amount of AST compound present in the sample.

Alanine aminotransferase (ALT) was determined by an enzymatic method based on studies by Wroblewski and LaDue [44], proposed as recommended by the International Federation of Clinical Chemistry, IFCC organization. In this reaction, the compound ALT catalyzes the transfer of an amino group from L-alanine to α-ketoglutarate to form L-glutamate and pyruvate. The change in absorbance difference for light wavelengths of 340 nm and 405 nm is due to the conversion of the NADH form to NAD+. It is directly proportional to the amount of ALT compound in the sample [45].

### 2.4. Statistic Methods

Compliance of analyzed quantitative variables with normal distribution was checked using the Shapiro–Wilk test. In order to check the differences in the studied parameters before and after taking the drug, the Wilcoxon test was applied. To check the differences between the study groups for changes in the studied parameters, Mann–Whitney U test was applied. McNemar’s test was also applied to check the differences for qualitative variables in the studied parameters before and after taking the drug. The level of statistical significance was taken at *p* < 0.05. STATISTICA 13.3 and PS IMAGO PRO 7 (IBM SPSS Statistics 27) were used for data analyses. Data are presented as a median with quartile deviation (Q).

## 3. Results

Recruitment began in 2015 and 2016 and ended in March 2020, and the intervention period also ended in March 2020. The flowchart of participants is shown in Figure 1. Of the 57 people who were assessed for eligibility, seven did not meet the inclusion criteria and were excluded from the study. A total of 50 participants were randomized to arm I (n = 25) and arm IOW (n = 25), received the allocated intervention, and completed the study. No major side effects were identified.

The baseline characteristics of the study population are shown in Table 2, and in most cases, no differences were observed between subjects assigned to arm I and arm IOW. Differences between groups were found only for HDL (*p* < 0.001) and LDL (*p* = 0.035) concentrations.

### 3.1. Evaluation of ALT, AST, TCH, LDL, HDL, and TG Parameters before and after Treatment with Isotretinoin or Isotretinoin Combined with Evening Primrose Oil

Before the study, all patients had normal ALT and AST test results that did not exceed the recommended reference values and showed no significant differences between groups I and IOW (AST *p* = 0.278, ALT *p* = 0.675). TCH, LDL, HDL, and TG levels were also within normal range in all patients in group I and IOW. For TC and TG values, no statistically significant differences were found between groups I and IOW (*p* > 0.05); however, a significantly higher median value for HDL was found in group I (*p* < 0.001) and for LDL in group IOW (*p* = 0.035).

After 9 months of isotretinoin treatment, patients showed a statistically significant increase in ALT (*p* < 0.001), AST (*p* < 0.001), TCH (*p* < 0.001), LDL (*p* < 0.001), and TG (*p* < 0.001), and a decrease in HDL (*p* = 0.013). The group treated with isotretinoin and supplemented with evening primrose oil after 9 months showed a statistically significant decrease in AST (*p* = 0.036), TCH (*p* = 0.025), and LDL (*p* = 0.003) levels, no change in ALT (*p* = 0.151), and an increase in HDL (*p* < 0.001) and TG (*p* = 0.025) levels. Significant differences between the isotretinoin-treated group and the isotretinoin combined with evening primrose group after 9 months were in AST (*p* = 0.001), TCH (*p* < 0.001), LDL (*p* = 0.003), and TG (*p* < 0.001) levels. In contrast, no differences were observed between the study groups for ALT (*p* = 0.217) and HDL (*p* = 0.366) concentrations. These results are presented in Table 3 and illustrated in Figure 2 and Figure 3.

The analysis of the distribution of subjects in groups for which the analyzed parameters were within or outside the norm before and after 9 months of treatment showed that, in group I, the percentage of subjects whose TCH levels exceeded the norm significantly increased from 8% before treatment to 84% after treatment (*p* < 0.001). For the other parameters (HDL, LDL, TG, AST, and ALT), no significant changes in distribution were observed in groups I and IOW. These results are summarized in Table 4.

### 3.2. Evaluation of Dietary Changes before and after Treatment with Isotretinoin or Isotretinoin Combined with Evening Primrose Oil

Before the treatments, no significant differences were found between the energy value of the studied rations of group I and IOW (*p* > 0.05), the content of protein, animal protein, vegetable protein, fat, carbohydrates, sucrose, and fiber, and the share of energy derived from these components (*p* > 0.05). There were also no differences between the contents of SFA (saturated fatty acids); MFA (monounsaturated fatty acids); EFA (essential fatty acids); n-3 and n-6 family fatty acids; oleic acid; gamma-linolenic acid and long-chain polyunsaturated fatty acids; vitamins A, E, C, B6, thiamine, and riboflavin; as well as iron, zinc, and copper in group I and IOW rations (for all *p* > 0.05).

After 9 months of treatment, no changes in nutritional parameters were observed in the isotretinoin-treated group (I) compared to the results obtained at the beginning of the study (*p* > 0.05) except for a decrease in mean vitamin E content (*p* = 0.028). However, in the group treated with isotretinoin combined with EPO supplementation, there was a significant decrease in the energy value of the diet (by 438 kcal, *p* = 0.005), a decrease in protein (*p* = 0.026), carbohydrates (*p* = 0.09), sucrose (*p* = 0.003), dietary fiber (*p* = 0.013), iron (*p* = 0.009), copper (*p* = 0.015), vitamin E (*p* = 0.042), riboflavin (*p* = 0.030), and vitamin B6 (*p* = 0.002). The niacin content of IOW rations also decreased by half (*p* = 0.002) and the vitamin C content decreased by two-thirds (*p* = 0.025). After 9 months of treatment, however, there were no significant differences between the groups for any of the parameters analyzed (*p* > 0.05 in all cases). There were also no significant differences for Δ before and after treatment between groups (*p* > 0.05 in all cases). These results are summarized in Table 5.

## 4. Discussion

Oral isotretinoin (13-cis-retinoic acid) was synthesized in 1955 [46] and was first approved for the treatment of severe forms of acne by the U.S. Food and Drug Administration (FDA) in 1982 [47]. To date, isotretinoin remains a highly effective treatment for moderate, severe, and recurrent acne, reliably leading to dermatological improvement after treatment [2,15,16,17,18,19,48]. Isotretinoin is the only therapy that affects all major etiologic factors of acne [47,49].

However, there are side effects that limit the use of isotretinoin. Among the most worrying are the suspected teratogenic effects [2]. Studies confirm that long-term high-dose use of retinoid-containing drugs can cause liver dysfunction without liver damage [23,50]. Hepatic dysfunction occurs in up to 15% of patients taking isotretinoin [22]. In a study by Schulpis et al. [51], a statistically significant increase in liver enzymes was observed after isotretinoin administration, and in a study by Nazarian et al. [52], a difficult-to-reduce increase in alanine aminotransaminase (ALT) was observed after 20 weeks of isotretinoin treatment. However, a recent study by Pona et al. showed that isotretinoin does not increase liver enzyme levels in patients with severe acne [53]. Numerous studies have observed significant, up to 5-fold, increases in transaminase activity from baseline values induced by isotretinoin administration [23,24,54]. In our study, isotretinoin treatment also significantly increased hepatic aminotransferases ALT and AST in group I. However, no increase in the concentration of hepatic aminotransferases was observed in the group treated with isotretinoin in combination with evening primrose oil; moreover, a decrease in AST levels was demonstrated after 9 months of treatment (*p* < 0.05).

Several studies have also observed an adverse effect of isotretinoin on changes in lipid profiles. In a study by Zech et al. [20], men with acne treated with oral isotretinoin for four months showed isotretinoin-induced increases in plasma TG and TCH levels, increased very-low-density lipoprotein (VLDL) cholesterol, increased LDL levels, and decreased HDL levels compared to pretreatment values. A study by De Marchi et al. [21] confirmed isotretinoin treatment-induced changes in plasma lipid concentrations. In ten acne patients treated with isotretinoin at a dose of 0.8 mg/kg body weight, TCH, LDL, and TG increased after 4 weeks of treatment. An increase in TCH, TG, and LDL and a decrease in HDL were also observed in 30 patients treated with 0.5 mg/kg isotretinoin [55]. In patients with PCOS complicated by severe cystic acne, isotretinoin treatment also significantly increased TG and TCH levels [36].

A meta-analysis by Lee et al. supported the effect of isotretinoin on lipid profiles, but in a small proportion of patients and without high risk [56]. In our study, the isotretinoin-treated group showed an increase in TCH and LDL levels and a decrease in HDL levels. The opposite results were observed in patients treated with isotretinoin and evening primrose oil; here, there was a decrease in TCH and LDL levels and an increase in HDL levels. TG was increased in both groups. Moreover, after 9 months of isotretinoin treatment in group I, the percentage of patients whose TCH levels exceeded the norm significantly increased from 8% before treatment to 84% after treatment. The improvement of lipid parameters and aminotransferase levels in patients treated with isotretinoin combined with evening primrose oil suggests a protective effect of this oil. The beneficial effect of evening primrose oil is related to its composition and the biological properties of its components, mainly linoleic acid (LA) and γ-linolenic acid (GLA), which belong to the group of omega-6 acids [26,27].

A very common adverse effect of isotretinoin therapy is hypertriglyceridemia [57]. In our study, an increase in TG concentration was observed in both study groups, but in the group treated with isotretinoin with evening primrose oil (IOW), TG concentration increased to a lesser extent than in the control group (I).

Analysis of the results regarding energy value and the content and proportion of energy derived from selected nutrients after 9 months of treatment with isotretinoin or isotretinoin combined with evening primrose oil supplementation provided interesting information. In the group treated with isotretinoin, only a decrease in vitamin E content was found, whereas in the group that additionally took evening primrose oil, the energy value of the diet and the content of protein, carbohydrates, sucrose, dietary fiber, iron, and copper as well as vitamins E, B2, B3, B6, and C decreased.

The lower energy value of the diet in subjects treated with isotretinoin combined with evening primrose oil may have influenced, to some extent, the favorable changes in the lipid profile. Holloszy and Fontana [58] showed lower serum TCH and LDL and higher HDL concentrations in patients with a lower energy diet compared to the group following a typical Western diet. Similarly, a study by Maciejewska et al. [59] found that a 30% reduction in dietary energy resulted in significantly lower TCH, LDL, and TG concentrations.

The observed reduction in dietary energy in the group treated with isotretinoin and evening primrose oil is probably related to the properties of this oil, which are known to help the body to better utilize fat [60] and increase insulin sensitivity [61].

It seems that the gamma-linolenic acid contained in evening primrose oil is mainly responsible for the decrease in food intake in the IOW group and the decrease in dietary energy value. There is no available literature on the effect of gamma-linolenic acid on appetite reduction, but there are reports indicating its positive influence on body weight reduction. In an animal model study, Takada et al. [62] observed a significant weight loss in animals fed a GLA supplemented diet compared to a control group. The weight loss effect associated with gamma-linolenic acid intake has also been observed in human models. In subjects experiencing significant weight loss after taking 890 mg GLA/day for 1 year, weight gain was significantly impaired again. This was due to GLA stimulating brown adipose tissue and increasing metabolic rates. In addition, the prostaglandins that formed initiated fat burning in the mitochondria of brown adipose tissue, which could be an effective factor in promoting weight loss [63]. Some studies have shown that gamma-linolenic acid (GLA) can increase resting metabolic rate (RMR), which may help with weight control. However, no effect on RMR or body mass index was observed in overweight young adults with a family history of obesity [29]. A study by Khamaisi et al. also showed insulin-sensitizing properties of GLA-alpha-lipoic acid conjugate associated with antioxidant activity [64]. Nevertheless, it is possible that evening primrose oil reduces intake through other mechanisms.

The decrease in carbohydrates observed in the IOW group also may have improved TCH, TG, HDL, and LDL levels [65,66,67]. This effect mainly relates to products containing simple carbohydrates [68,69] and their content in the diet of the IOW group significantly decreased after the 9-month treatment. However, the decrease in dietary carbohydrates was also accompanied by a decrease in dietary fiber, which has a protective effect on the lipid profile [70,71] and on ALT levels [72].

Reducing the other components in the diet had less significant effects on lipid profiles and aminotransferase activity. Interestingly, there was no reduction in fat or energy derived from fat in the diets of patients in the IOW group. There were also no significant changes in the fatty acid composition of the diet before and after treatment, which could have a significant effect on the changes in lipid profiles observed in this group.

However, iron and zinc decreased in the diets of IOW patients after 9 months of treatment. If and how iron affects lipid profiles and progression of atherosclerosis remains highly debatable. In the Mendelian study by Zhou et al., it was shown that genetically predicted higher blood iron levels are associated with disorders of lipid metabolism including hyperlipidemia and hypercholesterolemia. They are also consistently associated with lower blood concentrations of TCH and LDL [73]. Thus, the decrease in dietary iron after treatment with isotretinoin combined with evening primrose oil appears to be positive in terms of improving the lipid profile. On the other hand, a decrease in zinc content did not have a beneficial effect on lipid profile parameters. A comprehensive systematic review and meta-analysis conducted by Asbaghi et al. [74] showed that lipid profiles may be affected by zinc supplementation. In women with migraines, serum LDL and TCH levels also decreased after zinc supplementation [75]. Zinc supplementation also decreased ALT and AST levels in rats with nonalcoholic steatohepatitis [76]. Similarly, in patients with minimal hepatic encephalopathy, zinc supplementation significantly decreased ALT and AST levels [77].

It seems that the decrease in vitamins had no effect on improving lipid parameters and aminotransferase levels in IOW patients. A study by Ashor et al. showed no overall effect of vitamin C on TCH and HDL levels [78]. In addition, no improvement in TCH, LDL, and HDL levels was observed under the influence of vitamin E supplementation in exercising rats [79]. The greatest effect on lipid profiles among B vitamins was observed for vitamin B12. A clinical study in women showed that low B12 levels were associated with increased TCH, LDL, and TCH to HDL ratio [80]. However, for our study, no significant differences in vitamin B12 levels were found between groups and before and after treatment. Such differences were also not shown for vitamin D, for which observational studies confirm the association between its increased serum levels and a favorable lipid profile [81,82].

Taking into account the results of our study, it can be concluded that during isotretinoin treatment, evening primrose oil has a protective effect on changes in lipid profiles and transaminase activity. It also affects the energy value of the diet and the content of some of the components in it. However, longer studies are needed to make more reliable decisions about the use of evening primrose oil and its safety in clinical practice.

The study had some strengths and limitations. The main strengths of this study were the randomized controlled design, lasting 9 months. Furthermore, this is one of the first human studies comparing the effects of evening primrose oil in combination with isotretinoin and isotretinoin alone on TCH, LDL, HDL, TG, ALT, and AST levels. The main limitation of this study was the small group size, which was sufficient for inference. The lack of output homogeneity between arms I and IOW in terms of HDL and LDL concentrations was also a limitation. We also did not assess other factors that could potentially influence the results, such as alcohol consumption, smoking, and physical activity. However, all participants were instructed not to make changes in dietary habits or physical activity. Furthermore, the doses of linoleic acid and gamma-linolenic acid were taken from the information available from the evening primrose oil producer. Multiple testing adjustment was also not calculated, which could reduce the significance of differences in results obtained before and after treatment with isotretinoin or isotretinoin and evening primrose oil.

## Figures and Tables

**Figure 1 nutrients-14-01342-f001:**
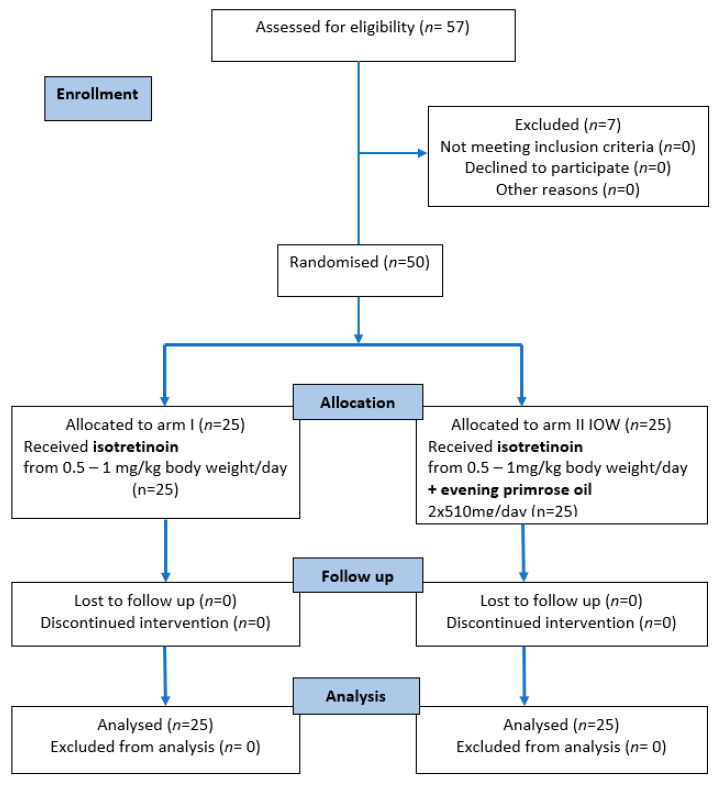
The CONSORT 2010 flow diagram.

**Figure 2 nutrients-14-01342-f002:**
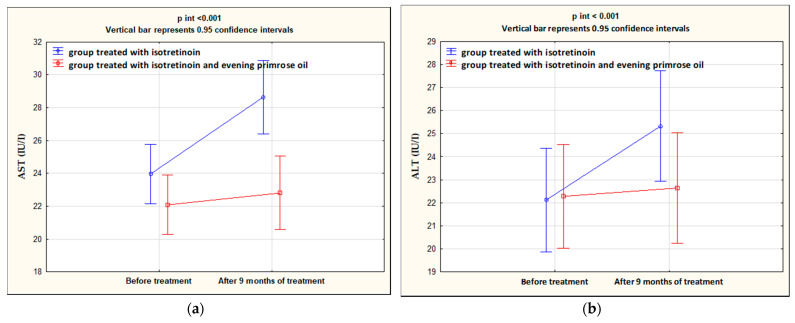
Changes in AST (**a**) and ALT (**b**) levels in the groups treated with isotretinoin or isotretinoin and evening primrose oil.

**Figure 3 nutrients-14-01342-f003:**
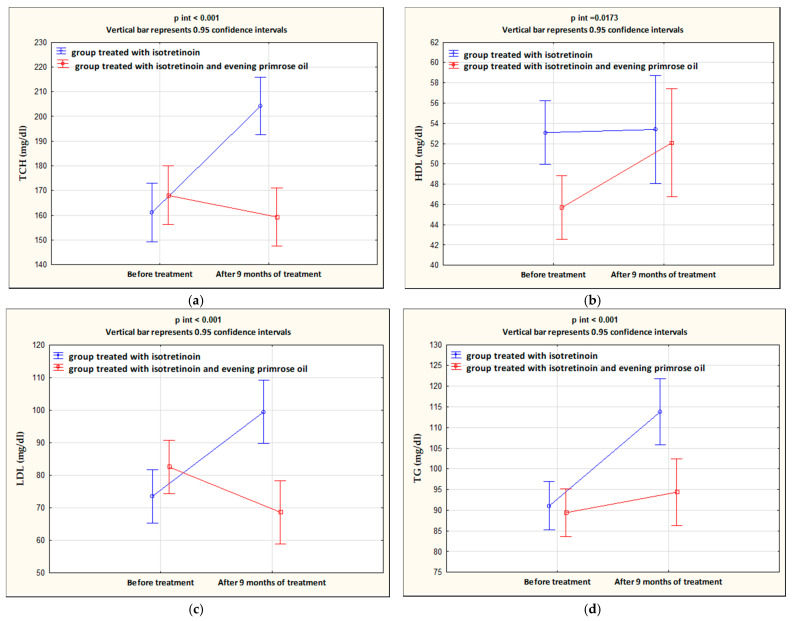
Changes in concentrations of TCH (**a**), HDL (**b**), LDL (**c**), and TG (**d**) in the groups treated with isotretinoin or isotretinoin and evening primrose oil.

**Table 1 nutrients-14-01342-t001:** Composition of evening primrose oil capsule.

Components	1 Capsule 42.8 g	4 Capsules 171 g
Evening primrose seed oil	510 mg	2040 mg
unsaturated fatty acids of which:	min. 390 mg	min. 1558 mg
linoleic acid (LA) gamma-linolenic acid (GLA)	min. 347 mg min. 42.4 mg	min. 1388 mg min. 170 mg

**Table 2 nutrients-14-01342-t002:** Baseline characteristics of the study population.

Analyzed Parameters	TOTAL RESEARCHED n = 50				Isotretinoin (I) n = 25		Isotretinoin with Evening Primrose Oil (IOW) n = 25		*p*
	X ± SD	Me ± Q	Min	Max	X ± SD	Me ± Q	X ± SD	Me ± Q	
Anthropometric parameters									
Body height (cm)	171 ± 5.79	169 ± 5.50	163	183	170 ± 5.79	169 ± 5.50	171 ± 5.79	169 ± 5.50	0.231
Body weight (kg)	65.0 ± 7.53	62.5 ± 4.50	55.0	86.0	64.6 ± 6.61	63.0 ± 4.00	65.4 ± 8.46	62.0 ± 5.00	0.953
BMI (kg/m^2^)	22.2 ± 1.42	22.0 ± 0.86	20.2	27.8	22.3 ± 1.15	22.3 ± 0.39	22.1 ± 1.67	21.8 ± 1.12	0.367
Biochemical parameters									
AST (IU/I)	23.0 ± 4.50	23.5 ± 2.00	12.0	36.0	24.0 ± 1.60	24.0 ± 1.00	22.1 ± 61.0	23.0 ± 4.50	0.278
ALT (IU/I)	22.2 ± 5.50	21.0 ± 3.00	9,0	35.0	22.1 ± 2.90	21.0 ± 2.00	22.3 ± 7.40	23.0 ± 6.00	0.675
TCH (mg/dL)	168 ± 21.1	168 ± 9.50	130	267	168 ± 23.1	163 ± 3.50	168 ± 19.4	171 ± 6.50	0.065
HDL (mg/dL)	49.9 ± 8.60	48.5 ± 5.05	35.0	82.0	531 ± 6.10	52.0 ± 2.50	45.7 ± 9.20	43.0 ± 3.50	<0.001
LDL (mg/dL)	78.0 ± 20.6	75.6 ± 14.2	42.6	150	73.6 ± 18.7	70.2 ± 6.75	82.5 ± 21.9	90.3 ± 17.1	0.035
TG (mg/dL)	90.2 ± 14.4	92.0 ± 6.50	52.0	130	91.0 ± 6.00	92.0 ± 1.00	89.4 ± 19.6	92.0 ± 9.00	0.892

n—Number of patients, X—mean, Me—median, Min—minimum value, Max—maximum value, SD—standard deviation, Q—quartile deviation, *p*—Mann–Whitney U test statistical significance value, ALT—alanine aminotransferase, AST—aspartate aminotransferase, TCH—total cholesterol, LDL—low-density lipoprotein, HDL—high-density lipoprotein, TG—triglycerides, BMI—body mass index.

**Table 3 nutrients-14-01342-t003:** Changes in ALT, AST, TCH, LDL, HDL, and TG levels or concentrations in groups using isotretinoin or isotretinoin with evening primrose oil.

Analyzed Parameters	Isotretinoin (I) n = 25			Wilcoxon Test	Isotretinoin with Evening Primrose Oil (IOW) n = 25			Wilcoxon Test	Mann–Whitney U Test for Differences between Groups I and IOW *p*		
	before Treatment	after 9 Months of Treatment	Δ	*p*	before Treatment	after 9 Months of Treatment	Δ	*p*	before Treatment	after Treatment	Δ before and after Treatment
	Me ± Q	Me ± Q	Me ± Q		Me ± Q	Me ± Q	Me ± Q				
AST (IU/I)	24.0 ± 1.00	28.0 ± 2.00	4.70 ± 4.20	<0.001	23.0 ± 4.50	22.0 ± 5.00	0.70 ± 3.80	0.036	0.278	0.001	<0.001
ALT (IU/I)	21.0 ± 2.00	24.0 ± 2.50	3.20 ± 1.30	<0.001	23.0 ± 6.00	22.0 ± 6.00	0.40 ± 3.80	0.151	0.675	0.217	<0.001
TCH (mg/dL)	163 ± 3.50	198 ± 5.00	35.9 ± 16.5	<0.001	171 ± 6.50	161 ± 9.00	−8.70 ± 16.1	0.025	0.065	<0.001	<0.001
HDL (mg/dL)	52.0 ± 2.50	51.0 ± 3.00	0.30 ± 6.30	0.013	43.0 ± 3.50	48.0 ± 5.00	6.40 ± 10.6	<0.001	<0.001	0.366	<0.001
LDL (mg/dL)	70.2 ± 6.80	95.9 ± 6.20	25.9 ± 12.6	<0.001	90.3 ± 17.1	60.2 ± 11.6	−13.9 ± 19.5	0.003	0.035	<0.001	<0.001
TG (mg/dL)	92.0 ± 1.00	114 ± 8.00	22.8 ± 9.50	<0.001	92.0 ± 9.00	95.0 ± 13.5	5.00 ± 15.7	0.025	0.892	<0.001	<0.001

n—Number of patients, Me—median, Q—quartile deviation, *p*—value of statistical significance, Δ—change, ALT—alanine aminotransferase, AST—aspartate aminotransferase, TCH—total cholesterol, LDL—low-density lipoprotein, HDL—high-density lipoprotein, TG—triglycerides.

**Table 4 nutrients-14-01342-t004:** Percentage of subjects in groups I and IOW for which the analyzed parameters were within or outside the reference ranges.

Analyzed Parameter	Reference Value	Isotretinoin(I) n = 25				McNemar’s Test	Isotretinoin with Evening Primrose Oil (IOW) n = 25				McNemar’s Test
		before Treatment		after Treatment			before Treatment		after Treatment		
		Normal [%]	Beyond the Norm [%]	Normal [%]	Beyond the Norm [%]		Normal [%]	Beyond the Norm [%]	Normal [%]	Beyond the Norm [%]	
TCH (mg/dL)	150–190 mg/dL	92	8	16	84	<0.001	92	8	96	4	1.000
HDL (mg/dL)	M—35–70 mg/dL K—40–80 mg/dL	100	0	100	0	No statistically significant changes	100	0	100	0	No statistically significant changes
LDL (mg/dL)	<115 mg/dL	96	4	96	4		100	0	100	0	
TG (mg/dL)	35–150 mg/dL	100	0	100	0		100	0	96	4	
AST (IU/I)	0–32 U/L	100	0	96	4		100	0	92	8	
ALT (IU/I)	0–33 U/L	100	0	100	0		100	0	100	0	

n—Number of patients, K—women, M—men, ALT—alanine aminotransferase, AST—aspartate aminotransferase, TCH—total cholesterol, LDL—low-density lipoprotein, HDL—high-density lipoprotein, TG—triglycerides.

**Table 5 nutrients-14-01342-t005:** Content of analyzed nutrients in whole daily rations of the study group before and after treatment with isotretinoin (I) or isotretinoin combined with evening primrose oil (IOW).

Analyzed Parameters	Isotretinoin (I) n = 25		Wilcoxon Test *p*	Isotretinoin with Evening Primrose Oil (IOW) n = 25		Wilcoxon Test *p*	Mann–Whitney U test For Differences between Groups I and IOW *p*		
	before Treatment	after 9 Months of Treatment		before Treatment	after 9 Months of Treatment		before Treatment	after 9 Months of Treatment	Δ before and after Treatment
	Me ± Q	Me ± Q		Me ± Q	Me ± Q				
Nutrients/Diet									
WE (kcal)	1678 ± 598	1540 ± 147	0.242	1747 ± 400	1308 ± 432	0.005	0.138	0.233	0.764
Protein (g)	83.1 ± 18.8	80.6 ± 8.10	0.264	73.7 ± 14.6	67.3 ± 16.3	0.026	0.282	0.491	0.479
Protein (% energy)	19.8 ± 3.98	20.9 ± 2.74	0.696	18.3 ± 3.27	19.9 ± 4.74	0.078	0.337	0.491	0.954
Animal protein (g)	51.7 ± 13.4	55.8 ± 6.37	0.242	49.4 ± 7.70	41.8 ± 17.8	0.150	0.648	0.930	0.691
Vegetable protein (g)	26.7 ± 7.68	24.8 ± 4.71	0.061	25.6 ± 6.78	19.1 ± 7.95	0.054	0.265	0.793	0.635
Fat (g)	38.8 ± 22.1	34.1 ± 9.56	0.326	32.4 ± 13.3	28.4 ± 13.9	0.201	0.233	0.177	0.900
Fat (% energy)	20.8 ± 5.39	18.4 ± 3.76	0.619	19.1 ± 4.10	23.5 ± 7.09	0.313	0.648	0.823	0.282
Carbohydrates (g)	236 ± 107	215 ± 53.2	0.192	256 ± 99.6	177 ± 81.9	0.009	0.177	0.154	0.580
Carbohydrates (% energy)	56.0 ± 6.76	54.6 ± 7.15	0.696	58.1 ± 5.44	53.1 ± 9.22	0.183	0.580	0.861	0.528
Saccharose(g)	18.6 ± 4.65	13.3 ± 3.66	0.300	19.6 ± 4.92	11.9 ± 9.75	0.003	0.443	0.327	0.282
Fiber (g)	14.4 ± 5.65	11.4 ± 7.05	0.619	20.4 ± 8.40	9.30 ± 8.86	0.013	0.056	0.367	0.097
Fatty acids									
SFA (g)	10.7 ± 6.88	9.30 ± 1.97	0.326	13.2 ± 5.29	10.7 ± 3.09	0.367	0.705	0.299	0.977
MFA (g)	9.70 ± 6.87	8.00 ± 3.95	0.493	12.0 ± 5.81	9.70 ± 4.40	0.313	0.265	0.160	0.930
NNKT (g)	3.90 ± 2.24	4.00 ± 1.15	0.563	5.10 ± 3.24	4.50 ± 2.03	0.201	0.265	0.699	0.992
n-3 (mg)	30.5 ± 0.23	0.50 ± 0.19	0.840	0.50 ± 0.14	0.50 ± 0.44	0.737	0.900	0.977	0.720
n-6 (mg)	63.5 ± 2.03	3.30 ± 0.92	0.288	4.00 ± 2.72	3.40 ± 1.79	0.083	0.299	0.992	0.930
Oleic acid n-9 (mg)	98.6 ± 6.11	7.10 ± 3.51	0.313	9.00 ± 5.83	8.90 ± 2.71	0.737	0.337	0.177	0.778
Gamma-linolenic acid (mg)	0.4 ± 0.18	0.5 ± 0.15	0.946	0.40 ± 0.15	0.40 ± 0.16	0.351	0.580	0.554	0.808
Long-chain polyunsaturated fatty acid	0.00 ± 0.03	0.00 ± 0.00	0.058	0.10 ± 0.04	0.00 ± 0.06	0.128	0.187	0.567	0.808
Mineral components									
Iron (mg)	8.10 ± 4.18	7.90 ± 3.20	0.716	13.5 ± 4.23	8.40 ± 4.26	0.009	0.076	0.421	0.720
Zink (mg)	6.60 ± 1.99	63.0 ± 0.73	0.264	8.20 ± 1.56	7.30 ± 2.08	0.128	0.184	0.290	0.691
Copper (mg)	0.90 ± 0.22	0.70 ± 0.22	0.893	0.90 ± 0.20	0.70 ± 0.34	0.015	0.109	0.677	0.290
Vitamins									
Vitamin A (µg)	793 ± 407	300 ± 261	0.716	614 ± 166	434 ± 179	0.276	0.635	0.720	0.648
Vitamin E (mg)	5.60 ± 2.20	3.40 ± 1.54	0.028	5.50 ± 2.31	3.70 ± 1.78	0.042	0.823	0.764	0.823
Thiamine (mg)	0.80 ± 0.51	0.70 ± 0.37	0.840	1.20 ± 0.48	1.20 ± 0.55	0.098	0.079	0.491	0.479
Riboflavin (mg)	1.20 ± 0.40	1.00 ± 0.27	0.427	1.30 ± 0.41	1.30 ± 0.32	0.030	0.337	0.930	0.443
Niacin (mg)	19.7 ± 8.33	15.2 ± 8.71	0.989	32.9 ± 13.4	15.4 ± 7.42	0.002	0.047	0.808	0.154
Vitamin B6 (mg)	1.20 ± 0.40	1.00 ± 0.27	0.989	2.22 ± 0.75	1.70 ± 0.60	0.002	0.051	0.900	0.211
Vitamin C (mg)	19.7 ± 8.33	15.2 ± 87.1	0.122	86.2 ± 36.3	27.2 ± 15.0	0.025	0.854	0.070	0.248
Vitamin B12 (µg)	2.20 ± 0.69	2.20 ± 0.53	0.677	2.30 ± 0.60	1.90 ± 1.08	0.288	0.594	0.691	0.290
Vitamin D (µg)	1.30 ± 1.18	1.00 ± 1.01	0.946	1.30 ± 0.88	1.80 ± 0.89	0.638	0.635	0.190	0.720

n—Number of patients, Me—median, Q—quartile deviation, *p*—statistical significance value, EC—energy value, SFA—saturated fatty acids, MFA—monounsaturated fatty acids, EFA—essential unsaturated fatty acids, C 18:3—gamma-linolenic acid.

## Data Availability

Data supporting the results obtained are deposited with the authors for review.

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
