# Peer review of "Effect of Evening Primrose Oil Supplementation on Biochemical Parameters and Nutrition of Patients Treated with Isotretinoin for Acne Vulgaris: A Randomized Double-Blind Trial"

_nutrients, 2022, doi:10.3390/nu14071342_

Round 1

Reviewer 1 Report

This study provides valuable data to demonstrate that the evening primrose oil has a beneficial effect during isotretinoin treatment. There are some comments for this manuscript.

Minor revisions:

  1. Line 22, (4x510 mg/day) is better than (4x510 mg).
  2. Line 124, value of food". [35]., the extra dot should be deleted.
  3. Line 231, TCH (p=0.000), its p-value is wrong.
  4. Line 33, “reducing energy requirements.” is not an appropriate description for the conclusion. Please revised the description.

Major revisions:

  1. The quality of the Figure 1 is not good enough, please revise it.
  2. The quality illustration of the evening primrose oil is required. Please provide qualitative and quantitative evidence for the oil used in this study, especially for the quantity of γ-linolenic acid.
  3. The first paragraph in the Results section is duplicated from the 2.1. Study participants of the Materials and Methods section. It is inappropriate, please revised it.

Author Response

RESPONSE TO REVIEWER 1 COMMENTS

We want to extend our appreciation to the reviewers and editorial board for taking the time and effort necessary to improve our work and provide such insightful guidelines.

We are very excited to have been given the opportunity to revise our manuscript which is entitled “Effect of evening primrose oil supplementation on biochemical parameters and nutrition of patients treated with isotretinoin for acne vulgaris: a randomized double-blind trial.”

We have carefully considered your comments. Herein we explain how we revised the paper based on your comments and recommendations.

We have highlighted the corrections and amended passages in red.

Below is a detailed response to the Reviewer:

REVIEWER

This study provides valuable data to demonstrate that the evening primrose oil has a beneficial effect during isotretinoin treatment. There are some comments for this manuscript.

Minor revisions:

  1. Line 22, (4x510 mg/day) is better than (4x510 mg).

Response: Thank you very much for your fair comment. This has been corrected.

REVIEWER

  1. Line 124, value of food". [35]., the extra dot should be deleted.

Response: Thank you very much for your fair comment. This has been corrected.

REVIEWER

  1. Line 231, TCH (p=0.000), its p-value is wrong.

Response: Thank you very much for noticing the error. It has been corrected.

The group treated with isotretinoin and supplemented with evening primrose oil after 9 months showed a statistically significant decrease in AST (p=0.036) and in TCH (p=0.025) and LDL (p=0.003) levels, no change in ALT (p=0.151) and an increase in HDL (p=0.000) and TG (p=0.025) levels.

REVIEWER

  1. Line 33, “reducing energy requirements.” is not an appropriate description for the conclusion. Please revised the description.

Response: Thank you very much for your valuable comment. The sentence has been corrected to:

The evening primrose oil treatment group also showed a reduction in dietary energy due to a reduction in dietary protein and carbohydrates.

REVIEWER

Major revisions:

  1. The quality of the Figure 1 is not good enough, please revise it.

Response: Thank you very much for your valuable comment. Figure 1 has been corrected according to CONSORT guidelines and moved to the results section.

REVIEWER

  1. The quality illustration of the evening primrose oil is required. Please provide qualitative and quantitative evidence for the oil used in this study, especially for the quantity of γ-linolenic acid.

Response: Thank you very much for your valuable comment.The evening primrose oil used in the study was described along with the linoleic and gamma-linolenic acid contents.

The encapsulated evening primrose oil used in the study was extracted from Oenothera paradoxa and manufactured by Adamed Consumer Healthcare S.A. The capsule contained evening primrose seed oil (Oenothera paradoxa) - 510 mg, unsaturated fatty acids - 390 mg including: linoleic acid (LA) - 347 mg, gamma-linolenic acid (GLA) - 42.4 mg, gelatin, humectants - glycerol, sorbitol. According to the manufacturer's recommendations, the recommended intake is not exceeded, which is for adolescents and adults 1-2 capsules 2 times a day.

Components

1 capsule

42,8 g

4 capsules

171 g

Evening Primrose Seed Oil

510 mg

2040 mg

unsaturated fatty acids

of which:

min. 390 mg

min. 1558 mg

linoleic acid (LA)

gamma-linolenic acid (GLA)

min. 347 mg

min. 42,4mg

min. 1388 mg

min. 170 mg

REVIEWER

  1. The first paragraph in the Results section is duplicated from the 2.1. Study participants of the Materials and Methods section. It is inappropriate, please revised it.

Response: Thank you very much for your valuable comment. The passage has been removed.

We also changed the order of references, which was corrected in the text and in the list of references. By introducing an additional table, the numbering of the remaining tables has also been changed.

On behalf of the entire team, I would like to thank you for your valuable suggestions, which gave the paper a completely different shape.

Yours sincerely

Izabela Bolesławska

Reviewer 2 Report

Firstly, the authors are invited to review the checklists of Consolidated Standards of Reporting Trials (CONSORT) and The Standards for Reporting Interventions in Controlled Trials of Acupuncture (STRICTA) statements. They are kindly requested to revise their manuscript accordingly.

What is the type of study?

The type of study should be a part of the title.

The abstract should be more organized and conclusive.

In which scientific base, the authors have enrolled moderate to severe acne vulgaris.

What is the clinical registration of this clinical trial?

Did the authors conduct power analyses for the sample size assessment? 

The discussion should be more organized.   It needs more scientific explanations.

Major corrections for the language are essential.

References should be matched with the instructions of this journal.

Author Response

RESPONSE TO REVIEWER 2 COMMENTS

We want to extend our appreciation to the reviewers and editorial board for taking the time and effort necessary to improve our work and provide such insightful guidelines.

We are very excited to have been given the opportunity to revise our manuscript which is entitled “Effect of evening primrose oil supplementation on biochemical parameters and nutrition of patients treated with isotretinoin for acne vulgaris: a randomized double-blind trial.”.

We have carefully considered your comments. Herein we explain how we revised the paper based on your comments and recommendations.

We have highlighted the corrections and amended passages in red.

Below is a detailed response to the Reviewer:

REVIEWER

  1. Firstly, the authors are invited to review the checklists of Consolidated Standards of Reporting Trials (CONSORT) and The Standards for Reporting Interventions in Controlled Trials of Acupuncture (STRICTA) statements. They are kindly requested to revise their manuscript accordingly.

Response: Thank you very much for the very valuable tip. We have read the indicated checklists and completed the manuscript with the necessary information. We have also included 2 items in the literature. We have included the CONSORT checklist in the supplementary material.

REVIEWER

  1. What is the type of study?

Response: Thank you very much for pointing out that we were not clear enough about the type of study. It was a randomised double-blind study.

REVIEWER

  1. The type of study should be a part of the title.

Response: Thank you very much for pointing out that we were not clear enough about the type of study. It was a randomised double-blind study.

Effect of evening primrose oil supplementation on biochemical parameters and nutrition of patients treated with isotretinoin for acne vulgaris: a randomized double-blind trial.

REVIEWER

  1. The abstract should be more organized and conclusive.

Response: Thank you very much for your valuable comment. The abstract has been corrected.

Abstract: Acne vulgaris is one of the most common skin diseases. One of the therapeutic options recommended for severe acne or acne that has not responded to previous therapies is isotretinoin dosage. However, its use may lead to adverse changes in the serum lipid profile and increased levels of transaminases. In this study we evaluated the effect of supplementation with evening primrose seed oil in acne vulgaris patients treated with isotretinoin on blood lipid parameters and transaminases activity. Study participants were randomly assigned to two treatments: conventional with isotretinoin (25 patients) and novel with isotretinoin combined with evening primrose seed oil (4x510 mg/day) (25 patients) for 9 months. Outcome variables included changes in total cholesterol (TCH), high-density lipoprotein (HDL) cholesterol, triglycerides (TG), LDL fraction cholesterol (LDL), alanine aminotransferase (ALT), aspartate aminotransferase (AST) and diet. Compared with isotretinoin treatment, the introduction of isotretinoin treatment combined with evening primrose oil had a positive effect on TCH concentrations (mean: 198 vs 161, p=0.000), LDL (95.9 vs 60.2, p=0.000), HDL (51.0 vs 48.0, p=0.000), TG (114 vs 95.0, p=0.000), ALT (24.0 vs 22.0, p=0.000) and AST (28.0 vs 22.0, p=0.000) but had no effect on the energy and ingredient content of the diets (p>0.005). Evening primrose oil was found to have beneficial effects on lipid profile and transaminase activity during isotretinoin treatment. However, longer studies are needed to make more reliable decisions regarding the use of evening primrose oil and its safety in clinical practice. The evening primrose oil treatment group also showed a reduction in dietary energy due to a reduction in dietary protein and carbohydrates.

REVIEWER

  1. In which scientific base, the authors have enrolled moderate to severe acne vulgaris.

Response: Thank you very much for your valuable question. We did not include moderate to severe acne vulgaris in any database because the study was not reported.

REVIEWER

  1. What is the clinical registration of this clinical trial?

Response: This study has no clinical registration. The results are with the study authors and can be viewed at any time.

REVIEWER

  1. Did the authors conduct power analyses for the sample size assessment? 

Response: Thank you very much for your valuable question and also your tip.

A power analysis was conducted prior to the study to assess the sample size. We have included this information in the text of the paper.

The minimum sample size was calculated using Statistica StatSoft 13.3 data analysis software package based on the following assumptions: a Type I probability of error of α = 0.05 and an allocation ratio of 1:1. As assumed, the minimum sample size was 23 subjects, but assuming a 10% dropout rate, we assumed that at least 25 subjects should be recruited for the study. We calculated the minimum sample size based on changes in TCH previously reported by Acmaz et al [1]. The power for this size is 0.9072.

REVIEWER

  1. The discussion should be more organized.   It needs more scientific explanations.

Response: Thank you very much for your valuable information. The discussion has been reorganised. Several scientific items have been added. Changes have also been made in the description of the limitations of the study.

REVIEWER

  1. Major corrections for the language are essential.

Response: Thank you very much for your attention. The work has been linguistically corrected by a native speaker.

REVIEWER

  1. References should be matched with the instructions of this journal.

Response: Thank you very much for your attention. References have been corrected according to the journal guidelines.

We also changed the order of references, which was corrected in the text and in the list of references. By introducing an additional table, the numbering of the remaining tables has also been changed.

On behalf of the entire team, I would like to thank you for your valuable suggestions, which gave the paper a completely different shape.

Yours sincerely

Izabela Bolesławska

Round 2

Reviewer 2 Report

Good work.

Author Response

Response to the Reviewer

We would like to thank you for taking the time and effort to improve our work and for providing the necessary guidance. They were very informative for our team. We will keep them in mind in the future. We are glad that we were able to improve the manuscript according to the reviewer's suggestions because it has acquired new values that we had previously overlooked.

We also thank you very much for your kind words.

On behalf of the team

Izabela Bolesławska
